# Concomitant Administration of Psychotropic and Prostate Cancer Drugs: A Pharmacoepidemiologic Study Using Drug–Drug Interaction Databases

**DOI:** 10.3390/biomedicines12091971

**Published:** 2024-09-01

**Authors:** Daniel Ungureanu, Adina Popa, Adina Nemeș, Cătălina-Angela Crișan

**Affiliations:** 1Department Pharmacy I, Discipline of Pharmaceutical Chemistry, “Iuliu Hațieganu” University of Medicine and Pharmacy, 41 Victor Babeș Street, 400012 Cluj-Napoca, Romania; daniel.ungureanu@elearn.umfcluj.ro; 2“Prof. Dr. Ion Chiricuță” Oncology Institute, 34-36 Republicii Street, 400015 Cluj-Napoca, Romania; 3Department Pharmacy II, Discipline of Clinical Pharmacy, “Iuliu Hațieganu” University of Medicine and Pharmacy, 12 Ion Creangă Street, 400010 Cluj-Napoca, Romania; 4Department of Oncology, Discipline of Medical Oncology, “Iuliu Hațieganu” University of Medicine and Pharmacy, 34-36 Republicii Street, 400015 Cluj-Napoca, Romania; 5Department of Neurosciences, Discipline of Psychiatry and Pediatric Psychiatry, “Iuliu Hațieganu” University of Medicine and Pharmacy, 43 Victor Babeș Street, 400012 Cluj-Napoca, Romania; ccrisan@umfcluj.ro; 6First Psychiatric Clinic, Emergency County Hospital, 43 Victor Babeș Street, 400012 Cluj-Napoca, Romania

**Keywords:** prostate cancer, psychotropic drug, drug–drug interaction, pharmacokinetic interactions, pharmacodynamic interactions, CYP3A4 induction, QT prolongation, drug interaction databases, pharmacoepidemiology

## Abstract

Prostate cancer (PC) represents the second most common diagnosed cancer in men. The burden of diagnosis and long-term treatment may frequently cause psychiatric disorders in patients, particularly depression. The most common PC treatment option is androgen deprivation therapy (ADT), which may be associated with taxane chemotherapy. In patients with both PC and psychiatric disorders, polypharmacy is frequently present, which increases the risk of drug–drug interactions (DDIs) and drug-related adverse effects. Therefore, this study aimed to conduct a pharmacoepidemiologic study of the concomitant administration of PC drugs and psychotropics using three drug interaction databases (Lexicomp^®^, drugs.com^®^, and Medscape^®^). This study assayed 4320 drug–drug combinations (DDCs) and identified 814 DDIs, out of which 405 (49.63%) were pharmacokinetic (PK) interactions and 411 (50.37%) were pharmacodynamic (PD) interactions. The most common PK interactions were based on CYP3A4 induction (*n* = 275, 67.90%), while the most common PD interactions were based on additive torsadogenicity (*n* = 391, 95.13%). Proposed measures for managing the identified DDIs included dose adjustments, drug substitutions, supplementary agents, parameters monitoring, or simply the avoidance of a given DDC. A significant heterogenicity was observed between the selected drug interaction databases, which can be mitigated by cross-referencing multiple databases in clinical practice.

## 1. Introduction

Worldwide, prostate cancer (PC) is the second most common diagnosed cancer in men and represents 7.3% of the new cancer cases diagnosed in 2022. The prognosis of patients diagnosed with PC is favorable, especially in the early stages, and therefore, despite its high incidence, the mortality rate of PC is not exceedingly high, where it represents only 3.8% of all cancer deaths [1,2,3]. The prostate is a male sex glandular structure and presents the highest rate of malignancy among the structures in the urogenital tract. Prostatic carcinogenesis is heavily reliant on androgenic hormone signaling, but it may also be linked to other signaling pathways like Sonic Hedgehog (Shh) expression and Gli-1 oncogene, for which the inappropriate expression consequently leads to tumoral growth and proliferation [4].

The burden of cancer diagnosis and long-term treatment with side effects on sexual function frequently cause depression in patients with PC, with many of them requiring psychological counseling and even psychiatric treatment that is either pharmacological or non-pharmacological [5,6]. Other signaled psychiatric comorbidities in patients diagnosed with PC include anxiety, post-traumatic stress disorder (PTSD), eating disorders, and schizophrenia [7,8,9,10,11]. Patients with PC are older patients with multiple associated comorbidities and have many drugs administered for these comorbidities. In addition, some of the drugs used in PC can cause psychiatric disorders as side effects, particularly depression, and that is why interdisciplinary collaboration with psychologists and psychiatrists is very important in patients with cancer, and the treatment of depression must be wisely chosen given the possible risk of drug–drug interactions (DDIs).

Androgen deprivation therapy (ADT) represents the standard treatment for patients with PC, either with localized disease in association with external beam radiation therapy (EBRT) or metastatic disease, as PC is an androgen-dependent disease. Androgens, in particular testosterone, stimulate the growth and survival of prostate cells and inhibit the apoptosis of prostate cells, whether normal or malignant. The main source of androgens is represented by the testis, but other sources of androgens can be involved in prostate cancer progression, especially in the castration-resistant setting, namely, the adrenal glands and the tumor cells themselves. ADT is based on the suppression of testis testosterone and can be achieved by surgical castration or medical castration with either luteinizing hormone-releasing hormone (LHRH) agonists or LHRH antagonists [12,13]. The use of LHRH antagonists (abarelix, degarelix, relugolix) seems very appealing due to the fact that their effect installs in less than 24 h, they show no flare effect, and the recovery of the pituitary–gonadal function is rapid after their discontinuation, but it is encumbered by serious histamine-mediated side effects and the lack of depot formulations beyond 1 month [14]. LHRH analogs (goserelin, triptorelin, leuprorelin/leuprolide) are widely preferred despite the flare-up phenomenon and possible side effects and represent the most frequent type of ADT. LHRH agonists are administered for 4–6 months up to 2–3 years in localized PC, in association with EBRT. In metastatic disease, maintaining the castration levels of testosterone is recommended no matter what other therapy is associated with the LHRH analogs, either androgen receptor pathway inhibitors (ARPIs) or other form of hormone therapy, chemotherapy, or targeted treatments [12,13,15].

LHRH agonists are associated with torsadogenic effects through QT prolongation, which can increase the risk of ventricular tachyarrhythmias, particularly torsade de pointes (TdP). The QT prolongation is caused by the T decline induced by LHRH agonists and other ADTs [16].

ARPIs (apalutamide, darolutamide, enzalutamide, abiraterone acetate) were proved to have major benefits in terms of overall survival (OS), progression-free survival (PFS), and quality of life in patients with non-metastatic castration resistant prostate cancer (CRPC) and metastatic prostate cancer, either hormone sensitive or castration resistant. The standard treatment in metastatic hormone sensitive prostate cancer (mHSPC) is the association between ADT and ARPI, or even triple therapy with the addition of docetaxel in fit patients with high-volume disease [12,13,15].

ARPI drugs can be classified into two types: steroidal (abiraterone acetate) and non-steroidal (enzalutamide, apalutamide, darolutamide, bicalutamide). Abiraterone acetate can influence the clearance of other drugs through the inhibition of cytochrome P450 isoenzymes (CYP) 2D6 and 2C8, and thus, affect the metabolism of approximately 25% of the co-administered drugs by increasing their plasmatic concentrations and the risk of adverse effects [17]. Among the potential adverse effects, it is important to note the proarrhythmogenic effects of abiraterone, which increase the risk of TdP [18].

Enzalutamide is a potent CYP2C9, 2C19, and 3A4 inducer that affects the metabolism of almost 50% of the possible co-administered drugs by lowering their plasmatic concentrations and efficacy [17]. According to the pharmacovigilance database Vigibase, enzalutamide was associated with QT prolongation similarly to abiraterone acetate [19].

Apalutamide is considered a strong inducer of CYP3A4 and 2C19 and a weak inducer of CYP2C9, with additional data suggesting that it may also induce the P-glycoprotein (P-gp) and the breast cancer resistance protein (BCRP) transporter [20]. Supplementarily, apalutamide was also associated with a modest QT-prolonging effect [21].

Darolutamide, according to in vitro evidence, is a BCRP transporter inhibitor and a potential CYP3A4 inducer [19,22]. Bicalutamide has the potential to inhibit CYP3A4 and may also be considered a weak CYP2C9, 2C19, and 2D6 inhibitor [23,24]. The QT prolongation potential of darolutamide and bicalutamide has not been clearly established, but literature data suggest that the torsadogenic effect of ARPIs may be considered a class effect, which occurs as a consequence of lowering testosterone levels [19].

Chemotherapy with docetaxel used to represent the treatment of choice when all means of hormone therapy were exhausted in patients with metastatic PC, but data published in recent years showed improved OS when docetaxel was administered upfront in early phases of the disease. In patients with de novo mHSPC with high-volume disease, the association of ADT, docetaxel, and either abiraterone or darolutamide improved the OS and PFS when administered as a first line. Cabazitaxel is a newer member of the taxanes family that has antitumoral effects in patients with CRPC and is effective when docetaxel therapy resistance is present [12,13,15].

Taxanes have no influence on any CYP450 isoenzyme, but they are substrates for CYP3A4 and 3A5 isoenzymes. Therefore, interaction with a CYP3A4 inhibitor may increase the severity of the myelosuppression associated with them [25,26]. The pharmacological classification of the drugs used in the treatment of PC is presented in Table 1.

PC and psychiatric disorders are conditions that often require polypharmacy to achieve a satisfying therapeutic response. While the definition of polypharmacy regarding the number of medications is variable, it is a well-known fact that an increased number of administered medications is associated with poor adherence and a higher risk of adverse effects and DDIs, especially in geriatric patients [27,28,29,30]. In addition to the previously presented potential of PC drugs to produce DDIs, based on their pharmacologic properties, it is important to mention that psychotropic drugs are also at high risk to interact with a large variety of co-administered drugs [31,32,33]. Therefore, a medication review and reconciliation are a must in this situation to apply the best decisions based on the benefit/risk ratio and to avoid supplementary harm of the patients.

In this regard, the aim of this study was to conduct a pharmacoepidemiologic study of the concomitant administration of PC and psychotropic drugs by using three different DDI databases. This study identified the possible DDIs that can occur in clinical practice and are presented based on their prevalence and stratified by their mechanism and severity. Several recommendations were made regarding the best decisions that should be made in clinical practice to avoid potential harmful effects of the identified DDIs on patients.

## 2. Materials and Methods

The DDIs were collected from three DDI databases: Lexicomp^®^ (UpToDate, Waltham, MA, USA), drugs.com^®^ Drug Interaction Checker (Drugsite Trust, Auckland, New Zealand), and Medscape Drug Interaction Checker^®^ (WebMD, New York City, NY, USA), between April and June 2024 [34,35,36]. The risk rating and severity assessment for each database are presented in Table 2. It is important to note that there is no correlation between the risk rating and severity based on how they were presented in Table 2.

The results were stored in a local database using Microsoft Excel for Microsoft 365 MS0 version 2408 build 16.0.17928.20114 (Microsoft, Redmond, WA, USA) and sorted by the mechanism and severity (only for Lexicomp^®^ and drugs.com^®^ databases) of the DDIs. The severity of the interactions was ranked as follows: minor, moderate, and major.

The selected psychotropic drugs for evaluation were monoaminoxidase inhibitor antidepressants (MAOIs—iproniazid, phenelzine, tranylcypromine, isocarboxazid, clorgilline, and moclobemide), tricyclic and tetracyclic antidepressants (TCAs—amitriptyline, imipramine, trimipramine, clomipramine, nortriptyline, desipramine, tianeptine, and maprotiline), selective serotonin reuptake inhibitor antidepressants (SSRIs—fluoxetine, fluvoxamine, paroxetine, sertraline, citalopram, and escitalopram), noradrenaline and serotonin reuptake inhibitor antidepressants (NSRIs—venlafaxine, duloxetine, milnacipran, levomilnacipran, and desvenlafaxine), noradrenaline and dopamine reuptake inhibitor antidepressants (NDRIs—bupropion), α_2_-adrenergic antagonist and serotonin reuptake inhibitor antidepressants (AASRIs—mirtazapine and mianserin), selective noradrenaline reuptake inhibitor antidepressants (SNRIs—reboxetine and viloxazine), serotonin antagonist and reuptake inhibitor antidepressants (SARIs—trazodone and nefazodone), serotonin modulator and stimulator antidepressants (SMSs—vilazodone and vortioxetine), melatonin receptor agonist and serotonin receptor antagonist antidepressants (agomelatine), NDMA receptor antagonist antidepressants (esketamine), phenothiazine antipsychotics (chlorpromazine, methotrimeprazine or levomepromazine, promazine, triflupromazine, mesoridazine, thioridazine, fluphenazine, perphenazine, proclorpherazine, and trifluoperazine), thioxanthene antipsychotics (chlorprothixene, clopenthixol, flupentixol, thiothixene, and zuclopenthixol), butyrophenone antipsychotics (haloperidol, droperidol, benperidol, bromperidol, moperone, pipamperone, spiperone, timiperone, trifluperidol, melperone, and lumateperone), diphenylbutylpiperidine antipsychotics (fluspirilene, penfluridol, and pimozide), benzamide antipsychotics (amisulpride, levosulpiride, nemonapride, sulpiride, sultopride, and tiapride), tricyclic and tetracyclic antipsychotics (carpipramine, clocapramine, loxapine, clozapine, asenapine, olanzapine, quetiapine, and zotepine), benzoisoxazole and benzoisothiazole antipsychotics (iloperidone, lurasidone, paliperidone, risperidone, ziprasidone, and perospirone), phenylpiperazine and quinolone antipsychotics (aripiprazole, brexpiprazole, and cariprazine), miscellaneous antipsychotics (blonanserin, pimavanserin, and sertindole), benzodiazepines (bromazepam, chlordiazepoxide, cinolazepam, clonazepam, clorazepate, diazepam, flunitrazepam, loflazepate, lorazepam, medazepam, nitrazepam, oxazepam, prazepam, temazepam, alprazolam, triazolam, and midazolam), Z hypnotics (zolpidem, zopiclone, eszopiclone, and zaleplon), orexin receptor antagonists (suvorexant, daridorexant, and lemborexant), melatonin receptor agonists (ramelteon and tasimelteon), barbiturates (amobarbital, butalbital, cyclobarbital, pentobarbital, phenobarbital, secobarbital, and thiopental), azapirones (buspirone), gabapentinoids (gabapentin, pregabalin, phenibut, and baclofen), mood stabilizers (lithium, carbamazepine, valproic acid, lamotrigine, and topiramate), and stimulants (atomoxetine, modafinil, armodafinil, methylphenidate, dexmethylphenidate, serdexmethylphenidate, amphetamine, levoamphetamine, dextroamphetamine, lisdexamfetamine, and methamphetamine).

The selected prostate cancer drugs for evaluation were LHRH agonists (goserelin, triptorelin, and leuprolide or leuprorelin), steroidal and non-steroidal ARPIs (abiraterone, enzalutamide, apalutamide, darolutamide, and bicalutamide), and taxanes (docetaxel and cabazitaxel).

The interaction checking was performed for each psychotropic drug with each prostate cancer drug, which limited the analysis to no more than two drugs at a time.

The curated data were used to determine the number and percentages of DDIs and to stratify the results per databases, classes and subclasses of drugs, and mechanisms.

## 3. Results

A total of 4320 drug–drug combinations (DDCs) between 144 psychotropic drugs and 10 PC drugs were checked for DDIs on all three selected databases, which meant 1440 checked DDCs per database. Not all selected psychotropic drugs were indexed in the selected databases: 24 were absent from Lexicomp^®^, 34 from drugs.com^®^, and 40 from Medscape^®^. The complete database of this study is available in the Appendix A.

From the grand total, 1296 (30.00%) of the checked DDCs were between psychotropic drugs and LHRH agonists, 2160 (50.00%) were between psychotropics and ARPIs, and 864 (20.00%) were between psychotropic drugs and taxanes. There were 816 identified DDIs (18.89%), 2434 (56.34%) DDCs with no DDIs, and 1070 (24.77%) DDCs that could not be checked because some psychotropic drugs were not indexed in the selected databases (N/A) (Table 3).

Out of the 816 identified DDIs, 205 (25.12%) of them were found using Lexicomp^®^, 416 (50.98%) using drugs.com^®^, and 195 (23.90%) using Medscape^®^. The identified DDIs were classified by mechanism and severity (only for DDIs identified with Lexicomp^®^ and drugs.com^®^).

Based on the mechanism of interaction, 405 (49.63%) DDIs were identified as pharmacokinetic (PK) interactions and 411 (50.37%) as pharmacodynamic (PD) interactions.

The identified PK interactions involved the distribution and metabolism processes. Out of all 405 PK interactions, 245 (67.90%) were through CYP3A4 induction, 77 (19.01%) through CYP2D6 inhibition, 42 (10.37%) through CYP3A4 inhibition, 6 (1.48%) through CYP2C19 induction, 2 (0.49%) through CYP2D6 induction, 1 (0.25%) through CYP2C9 induction, 1 (0.25%) through P-gp induction, and 1 (0.25%) through BCRP transporter inhibition.

Out of all 411 PD interactions, 391 (95.13%) were through additive torsadogenic effects, 12 (2.92%) through antagonism, 6 (1.46%) through additive myelotoxicity, and 2 (0.49%) through additive epileptogenic effects.

Based on the severity of DDIs, the results were sorted per database. For the Lexicomp^®^ database, out of all 205 identified DDIs, 65 (31.71%) were classified as minor, 114 (55.61%) as moderate, and 26 (12.68%) as major. For the drugs.com^®^ database, out of all 416 identified DDIs, 7 (1.68%) were classified as minor, 298 (71.63%) as moderate, and 111 (26.69%) as major (Figure 1).

Regarding sorting the results per screened PC drug, 248 (30.39%) of the identified DDIs contained a LHRH agonist, 521 (63.85%) contained an ARPI and 47 (5.76%) contained a taxane. On the other hand, regarding sorting the results per psychotropic drug, 238 (28.56%) of the identified DDIs contained an antidepressant, 358 (43.87%) contained an antipsychotic, 150 (18.38%) contained a sedative-hypnotic or anxiolytic, 32 (3.92%) contained a mood stabilizer, and 43 (5.27%) contained a stimulant (Figure 2).

The analysis of identified DDIs also included the prevalence of DDIs between the classes of psychotropic and PC drugs. ARPI–antipsychotic (23.41%), LHRH agonist–antipsychotic (19.61%), ARPI–antidepressant (18.51%), ARPI–sedative (15.93%), and LHRH agonist–antidepressant (8.94%) accounted for the highest prevalences of DDIs. The results are presented in Table 4 and graphically represented in Figure 3.

An extended analysis of the DDCs with the highest prevalence of identified DDIs was conducted and the results are presented in Table 5, Table 6, Table 7, Table 8, Table 9 and Table 10. The analyzed combinations were ARPI–antipsychotic, LHRH agonist–antipsychotic, ARPI–antidepressant, ARPI–sedative-hypnotic or anxiolytic, and LHRH agonist–antidepressant, which accounted for 86.40% of all identified DDIs. The identified DDIs were sorted per PC drug and subclass of psychotropic drug, number of interactions, and mechanisms.

## 4. Discussion

### 4.1. Databases Overview

The selected DDI databases were chosen based on their availability. Lexicomp^®^ was accessible through institutional access, while drugs.com^®^ and Medscape^®^ were free access databases.

There was a large observed heterogeneity between the databases. The highest number of interactions was identified using drugs.com^®^, followed by Lexicomp^®^, and then Medscape^®^. There were many cases in which a DDI was detected only by one or two of the selected databases and much less frequently when it was available in all three databases. Similar observations regarding the discrepancies between different DDI databases were reported by other authors too [37,38,39].

Another identified issue was the absence of some psychotropic drugs in these databases. The highest number of absent drugs was in drugs.com^®^, where 43 psychotropic drugs were not indexed, shortly followed by Medscape^®^. One of the most important drugs that was not indexed was tiapride, which is included in alcohol withdrawal disorder therapeutical algorithms [40].

An attempt was made to stratify the DDIs based on their severity for all three databases. However, because the severity levels were not available in the Medscape^®^ database (Table 1), this was performed only for the other two databases. The results showed that both Lexicomp^®^ and drugs.com^®^ databases classified most of the DDIs as moderate, while the drugs.com^®^ databases tended to not index minor DDIs, in contrast with Lexicomp^®^.

It is worth mentioning that the observed severities could not have been correlated with a particular risk rating. For example, there were DDIs with major severity, but the risk rating was monitor closely, or moderate DDIs for which the recommended management was to generally avoid the combination.

Taking into consideration these observations, a possible option for the clinicians is to concomitantly use two or more DDI databases when performing a medication review.

### 4.2. Identified DDIs Overview

By analyzing the identified DDIs, over half were of moderate severity and there were almost equal percentages for both PK and PD mechanisms, with the latter being more prevalent. It is important to note that in a few cases, the same interaction had the same severity in all three databases, hence another piece of evidence of the observed heterogeneity.

Most of the DDIs were identified in the DDCs containing ARPIs and LHRH agonists as PC drugs and antipsychotics, antidepressants, and sedative-hypnotics or anxiolytics as psychotropic drugs. The number of identified DDIs for these classes was directly dependent on the number of compounds in a class. Therefore, ARPIs and antipsychotics accounted for the highest number of DDIs. The lowest number of DDIs was identified for taxanes and mood stabilizers.

The identified DDIs were homogenous in number between the representatives of LHRH agonists and taxanes. However, an evident difference was observed between ARPIs, where DDIs that contained darolutamide or bicalutamide were barely indexed in the Lexicomp^®^ and Medscape^®^ databases. The number of DDIs in the drugs.com^®^ database for darolutamide and bicalutamide was consistent with the number of DDIs for the other ARPIs.

Another observation was that in some cases, there were different DDIs identified for the same DDC. For example, for the abiraterone–amitriptyline combination, two databases identified PK interactions (CYP2D6 inhibition), while the other one identified a PD interaction (additive torsadogenicity). Rationally, the additive tosadogenecity may be a consequence of the CYP2D6 inhibition caused by abiraterone, which increases the plasmatic concentrations of amitriptyline, which is a known torsadogenic drug [41]. However, this mechanism was not presented in the description of DDI. This advocates for the previous recommendation in which the clinicians should be encouraged to use multiple DDI databases at the same time.

### 4.3. Pharmacokinetic Interactions

#### 4.3.1. CYP3A4 Induction

The CYP3A4 isoenzyme accounts for the highest proportion of metabolized drugs since it is the most expressed CYP450 isoenzyme. It is important to note that CYP3A isoenzymes and P-glycoprotein are expressed together on the same cells and, therefore, the DDIs that imply one of these two mechanisms should mention both of them [42,43].

As expected, the PK interactions were predominantly based on CYP3A4 induction, which, in most cases, affected the psychotropic drugs involved in this type of DDIs. The main clinical consequence was the decrease in plasma concentrations of the psychotropics and reduced efficacy. The most potent CYP3A4 inducers PC drugs were ARPI enzalutamide and apalutamide [19,20].

The exception to the rule was when phenobarbital, carbamazepine, or modafinil were associated with PC drugs, which are known CYP3A4 inducers and may decrease the plasmatic concentrations of the associated PC drugs [44,45].

General clinical management solutions include the dose increase of the psychotropic drugs if they are associated with enzalutamide or apalutamide; the dose increase of the PC drugs if they are associated with carbamazepine, phenobarbital, or modafinil; and the substitution with drugs that have a limited effect on the CYP3A4 isoenzyme if appropriate. One such example is switching carbamazepine with oxcarbazepine [46,47,48].

#### 4.3.2. CYP2D6 Inhibition

The second most frequent PK mechanism was based on the CYP2D6 inhibition, which was identified in the DDIs where abiraterone acetate was associated. CYP2D6 is an isoenzyme that can metabolize lipophilic nitrogenous bases, which include most of the psychotropic drugs currently used in therapy [49].

Therefore, this association may increase the plasmatic concentrations of psychotropics and enhance the possible adverse effects associated with these drugs, such as central nervous system (CNS) sedation, orthostatic hypotension, and respiratory depression.

The most appropriate solution is the monitoring for adverse effects and lowering the doses of the affected psychotropic drugs [50].

#### 4.3.3. CYP3A4 Inhibition

The inhibition of CYP3A4 as a PK mechanism was less frequent and associated with the DDCs containing bicalutamide and/or nefazodone and to a lesser extent fluvoxamine, viloxazine, and iloperidone [23,51,52,53,54]. Based on the DDC, the outcome of this DDI is an increase in the plasmatic concentrations of either PC or psychotropic drugs.

Solutions include dose adjustments of the affected drug or switching to an alternative drug with no effect on the CYP3A4 isoenzyme, like switching nefazodone to trazodone [55].

#### 4.3.4. CYP2C19 Induction

CYP2C19 induction was reported only between apalutamide and some TCA (imipramine, trimipramine, and clomipramine) and SSRI antidepressants (sertraline, citalopram, and escitalopram) and resulted in a decreased efficacy of the antidepressants.

It is important to note that only TCAs that contained a tertiary amine were subjected to this DDI, but no antidepressants that contained a secondary amine (nortriptyline, desipramine, and maprotiline) had this DDI. This is because the *N*-demethylation process, through which the active metabolites containing a secondary amine are obtained, is CYP2C19-mediated and has a higher impact on the parent compounds with tertiary amines [56,57]. No DDI was reported in the case of amitriptyline. However, based on the structural similarities between amitriptyline and the other TCAs, we believe that it should have been indexed as a DDI. This is also supported by the literature data, which states that amitriptyline is metabolized to nortriptyline through CYP2C19 [57].

Another important observation regarding PK interactions based on CYP2C19 induction is that they were indexed only in the Medscape^®^ database for the given antidepressants. This reinforced the observed heterogeneity between the selected DDI databases and further advocated for a harmonization between them.

#### 4.3.5. CYP2D6 and CYP2C9 Induction

These two types of PK interactions were reported in very few situations. The CYP2D6 induction was reported by the drugs.com^®^ database only between DDCs of vortioxetine with enzalutamide and apalutamide [35]. The literature data suggest that both ARPIs can act as CYP2D6 inducers, and hence, affect vortioxetine and lower its plasmatic concentrations [58]. However, this type of PK interaction should have been more frequently detected since most psychotropic drugs are CYP2D6 substrates, as stated previously.

CYP2C9 induction was reported only in a distinct case between apalutamide and fluoxetine, as indexed by the Medscape^®^ database [36].

#### 4.3.6. P-Glycoprotein Induction and BCRP Transporter Inhibition

Only one DDI that solely involved the P-gp induction was reported between docetaxel and trazodone and indexed in the Medscape^®^ database [36]. As previously mentioned, CYP3A4 and P-gp act synergistically and are both influenced by the same induction or inhibition processes [42,43]. Hence, the number of PK interactions that involve P-gp should be at least equal to the number of those that involve CYP3A4.

The sole DDI that involved the inhibition of a BCRP transporter was reported between darolutamide and lurasidone and indexed in the drugs.com^®^ database. Since both compounds are known as BRCP transporter inhibitors, this may be the reason why it was reported as an interaction only between them. However, additional literature data evidenced that clozapine and risperidone are also BCRP transporter inhibitors. The clinical consequences of this interaction are not clear since it is unknown how this transporter facilitates the penetration of psychotropic drugs through the brain–blood barrier (BBB) [59,60].

### 4.4. Pharmacodynamic Interactions

#### 4.4.1. Additive Torsadogenicity

The risk of TdP is dependent on numerous factors, including old age, genetic variants, concomitant cardiovascular diseases, dyselectrolytemia (particularly hypokalemia and hypocalcemia), and concomitant usage of loop diuretics, laxatives, and other torsadogenic drugs [61].

Additive torsadogenicity was the most commonly identified PD interaction, with a little over 95% of all PD interactions. This percentage may be biased, as not all drugs implied in this interaction have the same risk of QT prolongation. Therefore, another instrument may be necessary to correctly assess the risk of TdP for the selected drugs in this study. According to the QTdrugs List by CredibleMeds, the TdP risk of a given drug can be stratified into three categories: (a) known risk of TdP (the risk exists even when the drug is administered as recommended); (b) possible risk of TdP (they lack evidence of TdP risk when administered as recommended, but are known to cause QT prolongation); and (c) conditional risk of TdP (the risk is only under certain conditions, as discussed previously) [62].

The risk of TdP classified based on QTdrugs lists for the selected drugs is presented in Table 11.

According to this list (Table 11), the only PC drugs that possessed a risk were leuprolide, abiraterone, apalutamide, and bicalutamide, where no drug prolonged the QT interval at normal doses. The databases, however, considered all LHRH agonists and ARPIs as torsadogenic drugs, while the literature evidence considers QT prolongation to be a class mechanism [19].

Considering these observations, we recommend using the QTdrugs List to evaluate the torsadogenic risk of a regimen since it can distinguish the risk between the representatives of the same class. As a general management solution, we recommend that combinations of PC drugs and psychotropics with a known or possible risk of TdP should be avoided as much as possible and only used for emergencies (for example, haloperidol in severe psychomotor agitation [63]).

DDCs with a conditional risk of TdP should be used with periodic monitoring of EKG, serum electrolytes, and medication reconciliation if there are loop diuretics or laxatives associated in the regimen (for example, initiating potassium and calcium supplementation, switching to thiazide or thiazide-like diuretics for sparring calcium, or associating potassium-sparring diuretics if loop diuretics are necessary).

#### 4.4.2. Antagonism

This PD interaction was reported only by the Lexicomp^®^ database between triptorelin with various antipsychotics (chlorpromazine, fluphenazine, perphenazine, trifluoperazine, thiothixene, haloperidol, pimozide, loxapine, asenapine, paliperidone, risperidone) and clomipramine, which was classified as a major interaction [34].

According to Lexicomp^®^, the mechanism of this PD interaction is based on the antagonism between PC drugs and hyperprolactinemic agents, which increase the prolactin levels and decrease the number of gonadotropin-releasing hormone receptors [34].

Hyperprolactinemia may be induced through dopamine antagonism (antipsychotics) or serotonin stimulation (clomipramine), while LHRH agonists have been documented to compensate the hyperprolactinemia present in microprolactinomas [64,65].

The absence of this interaction for other LHRH agonists or psychotropics that antagonize dopamine or increase serotonin levels represents a serious issue. The antagonism produced by this interaction decreases the efficacy of LHRH agonists and may facilitate PC progression.

Therefore, the combination of LHRH agonists and hyperprolactinemic psychotropics should be avoided. If necessary, prolactin levels should be periodically evaluated. Another option may be the substitution with drugs that have low or no effect on prolactin levels, such as aripiprazole, ziprasidone (switch from risperidone or paliperidone), SSRIs, or NSRIs [66,67].

#### 4.4.3. Additive Myelotoxicity

This PD interaction was reported by the Lexicomp^®^ and drugs.com^®^ databases between taxanes with promazine and clozapine. The DDIs indexed by drugs.com^®^ were rated as major, in contrast with those indexed by Lexicomp^®^ [34,35].

Myelotoxicity affects all three types of blood cells, while clozapine and possibly other antipsychotics have been associated with neutropenia and agranulocytosis [68,69,70]. If taxane chemotherapy and clozapine must be associated, a solution for the management of agranulocytosis is to add a granulocyte-colony-stimulating factor (filgrastim) in the regimen [71].

#### 4.4.4. Additive Epileptogenicity

This interaction was indexed only on drugs.com^®^ and reported between bupropion with enzalutamide and apalutamide [35].

The mechanism is based on the ability of enzalutamide and apalutamide to interact with the GABA_A_ receptor and lower the seizure threshold, while bupropion can produce seizures in a dose-related manner [72,73].

The suggested solution by drugs.com^®^ is to closely monitor the risk of seizures, while bupropion should be slowly titrated and should not exceed the maximum recommended dose [35].

## 5. Conclusions

In conclusion, our study exposed vulnerabilities and heterogeneity between the selected databases, which may delay a proper management of the drug-related side effects and DDIs, and therefore, increase the hospitalization of patients and risk of death.

The heterogeneity is based on the different numbers of identified DDIs, different criteria for severity and risk rating, different degrees of severity for the same DDI, and false negative results that were observed throughout this study. Due to these differences and to mitigate practice errors, it is strongly recommended that clinicians cross-reference several databases when evaluating the risk of DDIs in a patient regimen.

This study assayed 4320 DDCs in three different databases and identified 814 DDIs. The proportions of PK and PD interactions were almost equal, with PD interactions being more prevalent. The most common PK interactions were based on CYP3A4 induction, while the most common PD interactions were based on additive torsadogenicity.

Proposed measures for managing the identified DDIs included dose adjustments, drug substitutions, supplementary agents, parameters monitoring, or simply the avoidance of a given DDC.

Several limitations of this study were identified, including the inability to identify a correlation between the risk rating and severity assessment of the identified DDIs, which would have required an extended statistical analysis. Another limitation was the lack of a validated data curation method, thus only an empirical methodology was employed. Finally, it is worth mentioning that only three DDI databases to which we had access were used, where we excluded other relevant databases that needed a separate subscription, like IBM Micromedex^®^ Drug Interactions.

Nevertheless, this study aimed to advocate for the harmonization of DDI databases in order to reduce the time spent for cross-referencing and increase the efficacy in clinical practice.

## Figures and Tables

**Figure 1 biomedicines-12-01971-f001:**
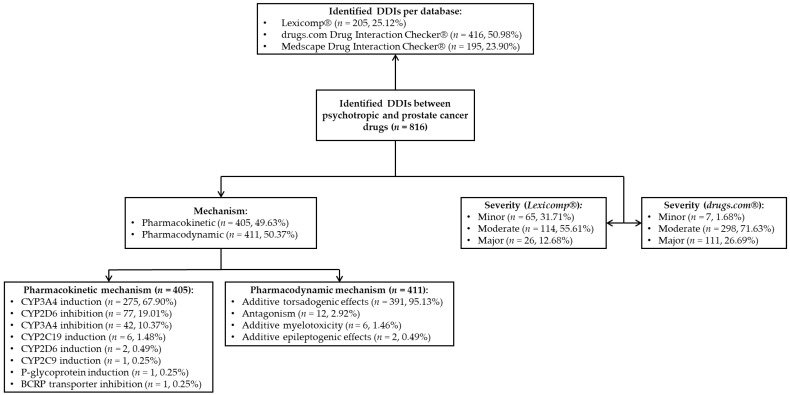
Classification of the identified DDIs from the selected databases.

**Figure 2 biomedicines-12-01971-f002:**
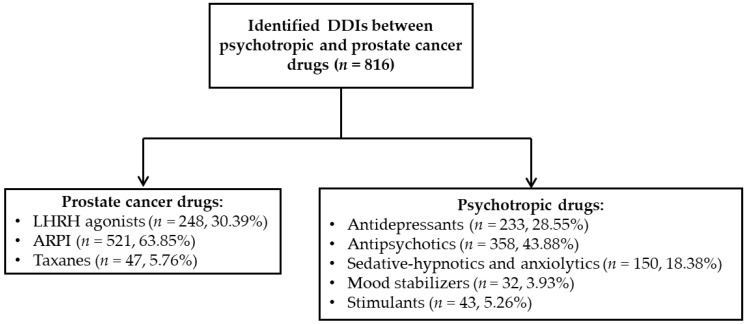
The distribution of identified DDIs based on the pharmacological class.

**Figure 3 biomedicines-12-01971-f003:**
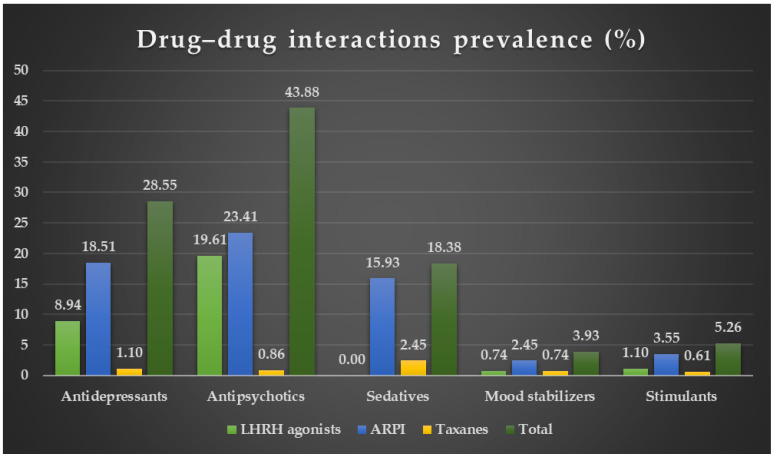
Graphical representation of the DDIs prevalence (%).

**Table 1 biomedicines-12-01971-t001:** Pharmacological classification of the drugs used in the treatment of PC [12,13,15].

LHRH Antagonists	LHRH Agonists	ARPIs	Taxanes
Abarelix	Goserelin	Abiraterone acetate	Docetaxel
Degarelix	Triptorelin	Apalutamide	Cabazitaxel
Relugolix	Leuprolide	Enzalutamide	
		Darolutamide	
		Bicalutamide	

**Table 2 biomedicines-12-01971-t002:** DDI risk rating and severity assessment for the selected databases (Lexicomp^®^, drugs.com^®^, and Medscape^®^) [34,35,36].

Lexicomp^®^		Drugs.com^®^		Medscape^®^
Risk Rating	Severity	Risk Rating	Severity	Risk Rating
Avoid combination	Major	Contraindicated	Major	Contraindicated
Consider therapy modification	Moderate	Generally avoiding	Moderate	Use alternative
Monitor therapy	Minor	Monitor closely	Minor	Monitor closely
No action needed				
No known interaction				

**Table 3 biomedicines-12-01971-t003:** The distribution of checked interactions per database. Legend: N/A—not available.

DDI Database	Identified DDIs (%)	N/A (%)	No Identified DDI (%)	Total (%)
**Lexicomp^®^**	205 (4.75)	240 (5.56)	994 (23.01)	1440 (33.34)
**drugs.com^®^**	416 (9.63)	430 (9.95)	595 (13.77)	1440 (33.33)
**Medscape^®^**	195 (4.51)	400 (9.26)	845 (19.56)	1400 (33.33)
**Total**	816 (18.89)	1070 (24.77)	2434 (56.34)	4320 (100.00)

**Table 4 biomedicines-12-01971-t004:** The prevalences of DDIs between the subclasses of psychotropic and PC drugs.

	LHRH Agonists (%)	ARPI (%)	Taxanes (%)	Total (%)
**Antidepressants**	73 (8.94)	151 (18.51)	9 (1.10)	233 (28.55)
**Antipsychotics**	160 (19.61)	191 (23.41)	7 (0.86)	358 (43.88)
**Sedative-hypnotics and anxiolytics**	0 (0.00)	130 (15.93)	20 (2.45)	150 (18.38)
**Mood stabilizers**	6 (0.74)	20 (2.45)	6 (0.74)	32 (3.93)
**Stimulants**	9 (1.10)	29 (3.55)	5 (0.61)	43 (5.26)
**Total**	248 (30.39)	521 (63.85)	47 (5.76)	816 (100.00)

**Table 5 biomedicines-12-01971-t005:** Analysis of the identified DDIs between ARPIs and antipsychotics.

DDCs	No. of DDIs (%) (*n* = 816)	Mechanism	PK/PD Mechanism
ARPI–Antipsychotic	191 (23.41)		
ARPI and phenothiazines	47 (5.76)	Pharmacokinetic (*n* = 11, 1.35)	CYP2D6 inhibition (*n* = 11, 1.35)
		Pharmacodynamic (*n* = 36, 4.41)	Additive torsadogenicity (*n* = 36, 4.41)
ARPI and benzisoxazoles or benzisothiazoles	36 (4.41)	Pharmacokinetic (*n* = 22, 2.69)	CYP3A4 induction (*n* = 15, 1.84)
			CYP2D6 inhibition (*n* = 5, 0.61)
			CYP3A4 inhibition (*n* = 1, 0.12)
			BCRP transporter inhibition (*n* = 1, 0.12)
		Pharmacodynamic (*n* = 14, 1.72)	Additive torsadogenicity (*n* = 14, 1.72)
ARPI and phenylpiperazines or quinolones	28 (3.43)	Pharmacokinetic (*n* = 26, 3.19)	CYP3A4 induction (*n* = 18, 2.21)
			CYP2D6 inhibition (*n* = 5, 0.61)
			CYP3A4 inhibition (*n* = 3, 0.37)
		Pharmacodynamic (*n* = 2, 0.25)	Additive torsadogenicity (*n* = 2, 0.25)
ARPI and tricyclics or tetracyclics	26 (3.19)	Pharmacokinetic (*n* = 11, 1.35)	CYP3A4 induction (*n* = 9, 1.10)
			CYP3A4 inhibition (*n* = 1, 0.12)
			CYP2D6 inhibition (*n* = 1, 0.11)
		Pharmacodynamic (*n* = 15, 1.84)	Additive torsadogenicity (*n* = 15, 1.84)

**Table 6 biomedicines-12-01971-t006:** Analysis of the identified DDIs between ARPIs and antipsychotics (continuation).

DDCs	No. of DDIs (%) (*n* = 816)	Mechanism	PK/PD Mechanism
ARPI–Antipsychotic	191 (23.41)		
ARPI and butyrophenones	26 (3.19)	Pharmacokinetic (*n* = 18, 2.21)	CYP3A4 induction (*n* = 14, 1.72)
			CYP3A4 inhibition (*n* = 2, 0.25)
			CYP2D6 inhibition (*n* = 2, 0.24)
		Pharmacodynamic (*n* = 8, 0.98)	Additive torsadogenicity (*n* = 8, 0.98)
ARPI and miscellaneous antipsychotics	13 (1.59)	Pharmacokinetic (*n* = 9, 1.10)	CYP3A4 induction (*n* = 8, 0.98)
			CYP2D6 inhibition (*n* = 1, 0.12)
		Pharmacodynamic (*n* = 4, 0.49)	Additive torsadogenicity (*n* = 4, 0.49)
ARPI and diphenylbutylpiperidines	8 (0.98)	Pharmacokinetic (*n* = 4, 0.49)	CYP3A4 induction (*n* = 2, 0.25)
			CYP3A4 inhibition (*n* = 1, 0.12)
			CYP2D6 inhibition (*n* = 1, 0.12)
		Pharmacodynamic (*n* = 4, 0.49)	Additive torsadogenicity (*n* = 4, 0.49)
ARPI and benzamides	4 (0.49)	Pharmacodynamic (*n* = 4, 0.49)	Additive torsadogenicity (*n* = 4, 0.49)
ARPI and thioxanthenes	3 (0.37)	Pharmacokinetic (*n* = 3, 0.37)	CYP3A4 induction (*n* = 2, 0.25)
			CYP2D6 inhibition (*n* = 1, 0.12)

**Table 7 biomedicines-12-01971-t007:** Analysis of the identified DDIs between LHRH agonists and antipsychotics.

DDCs	No. of DDIs (%) (*n* = 816)	Mechanism	PK/PD Mechanism
LHRH Agonist–Antipsychotic	160 (19.61)		
LHRH agonists and phenothiazines	45 (5.51)	Pharmacodynamic (*n* = 45, 5.51)	Antagonism (*n* = 4, 0.49)
			Additive torsadogenicity (*n* = 41, 5.02)
LHRH agonists and tricyclics or tetracylics	35 (4.29)	Pharmacodynamic (*n* = 35, 4.29)	Antagonism (*n* = 2, 0.25)
			Additive torsadogenicity (*n* = 33, 4.04)
LHRH agonists and benzisoxazoles or benzisothiazoles	28 (3.43)	Pharmacodynamic (*n* = 28, 3.43)	Antagonism (*n* = 2, 0.24)
			Additive torsadogenicity (*n* = 26, 3.19)
LHRH agonists and butyrophenones	18 (2.21)	Pharmacodynamic (*n* = 18, 2.21)	Antagonism (*n* = 1, 0.12)
			Additive torsadogenicity (*n* = 17, 2.09)
LHRH agonists and diphenylpiperidines	9 (1.10)	Pharmacodynamic (*n* = 9, 1.10)	Antagonism (*n* = 1, 0.12)
			Additive torsadogenicity (*n* = 8, 0.98)
LHRH agonists and benzamides	9 (1.10)	Pharmacodynamic (*n* = 9, 1.10)	Additive torsadogenicity (*n* = 9, 1.10)
LHRH agonists and phenylpiperazines or quinolones	6 (0.74)	Pharmacodynamic (*n* = 6, 0.74)	Additive torsadogenicity (*n* = 6, 0.74)
LHRH agonists and miscellaneous antipsychotics	6 (0.74)	Pharmacodynamic (*n* = 6, 0.74)	Additive torsadogenicity (*n* = 6, 0.74)
LHRH agonists and thioxanthenes	4 (0.49)	Pharmacodynamic (*n* = 4, 0.49)	Antagonism (*n* = 1, 0.12)
			Additive torsadogenicity (*n* = 3, 0.37)

**Table 8 biomedicines-12-01971-t008:** Analysis of the identified DDIs between ARPIs and antidepressants.

DDCs	No. of DDIs (%) (*n* = 816)	Mechanism	PK/PD Mechanism
ARPI–Antidepressant	151 (18.51)		
ARPIs and TCAs	49 (6.00)	Pharmacokinetic (*n* = 22, 2.70)	CYP2D6 inhibition (*n* = 19, 2.32)
			CYP2C19 induction (*n* = 3, 0.38)
		Pharmacodynamic (*n* = 27, 3.30)	Additive torsadogenicity (*n* = 27, 3.30)
ARPIs and SSRIs	41 (5.02)	Pharmacokinetic (*n* = 24, 2.94)	CYP2D6 inhibition (*n* = 10, 1.22)
			CYP3A4 induction (*n* = 8, 0.98)
			CYP2C19 induction (*n* = 4, 0.49)
			CYP3A4 inhibition (*n* = 2, 0.25)
		Pharmacodynamic (*n* = 17, 2.08)	Additive torsadogenicity (*n* = 17, 2.08)
ARPIs and SARIs	17 (2.08)	Pharmacokinetic (*n* = 13, 1.59)	CYP3A4 inhibition (*n* = 7, 0.86)
			CYP3A4 induction (*n* = 4, 0.49)
			CYP2D6 inhibition (*n* = 2, 0.25)
		Pharmacodynamic (*n* = 4, 0.49)	Additive torsadogenicity (*n* = 4, 0.49)
ARPIs and SMSs	14 (1.72)	Pharmacokinetic (*n* = 14, 0.49)	CYP3A4 induction (*n* = 10, 1.22)
			CYP2D6 inhibition (*n* = 2, 0.25)
			CYP2D6 induction (*n* = 2, 0.25)
ARPIs and NSRIs	13 (1.59)	Pharmacokinetic (*n* = 10, 1.22)	CYP2D6 inhibition (*n* = 5, 0.61)
			CYP3A4 induction (*n* = 4, 0.49)
			CYP3A4 inhibition (*n* = 1, 0.12)
		Pharmacodynamic (*n* = 3, 0.37)	Additive torsadogenicity (*n* = 3, 0.37)
ARPIs and AASRIs	12 (1.47)	Pharmacokinetic (*n* = 9, 1.10)	CYP3A4 induction (*n* = 8, 0,98)
			CYP2D6 inhibition (*n* = 1, 0.12)
		Pharmacodynamic (*n* = 3, 0.37)	Additive torsadogenicity (*n* = 3, 0.37)
ARPIs and SNRIs	3 (0.37)	Pharmacokinetic (*n* = 3, 0.37)	CYP3A4 induction (*n* = 2, 0.25)
			CYP3A4 inhibition (*n* = 1, 0.12)
ARPIs and NDRIs	2 (0.25)	Pharmacodynamic (*n* = 2, 0.25)	Additive epileptogenicity (*n* = 2, 0.25)

**Table 9 biomedicines-12-01971-t009:** Analysis of the identified DDIs between ARPIs and sedative-hypnotics or anxiolytics.

DDCs	No. of DDIs (%) (*n* = 816)	Mechanism	PK/PD Mechanism
ARPI–Sedative-Hypnotic or Anxiolytic	130 (15.93)		
ARPIs and benzodiazepines	45 (5.51)	Pharmacokinetic (*n* = 45, 5.51)	CYP3A4 induction (*n* = 41, 5.02)
			CYP3A4 inhibition (*n* = 3, 0.37)
			CYP2D6 inhibition (*n* = 1, 0.12)
ARPIs and orexin receptor antagonists	25 (3.06)	Pharmacokinetic (*n* = 25, 3.06)	CYP3A4 induction (*n* = 18, 2.21)
			CYP3A4 inhibition (*n* = 7, 0.86)
ARPIs and barbiturates	23 (2.82)	Pharmacokinetic (*n* = 23, 2.82)	CYP3A4 induction (*n* = 23, 2.82)
ARPIs and Z hypnotics	19 (2.33)	Pharmacokinetic (*n* = 19, 2.32)	CYP3A4 induction (*n* = 18, 2.20)
			CYP3A4 inhibition (*n* = 1, 0.12)
ARPIs and melatonin receptor agonists	11 (1.35)	Pharmacokinetic (*n* = 11, 1.35)	CYP3A4 induction (*n* = 10, 1.23)
			CYP3A4 inhibition (*n* = 1, 0.12)
ARPIs and azapirones	6 (0.74)	Pharmacokinetic (*n* = 6, 0.74)	CYP3A4 induction (*n* = 6, 0.74)

**Table 10 biomedicines-12-01971-t010:** Analysis of the identified DDIs between LHRH agonists and antidepressants.

DDCs	No. of DDIs (%) (*n* = 816)	Mechanism	PK/PD Mechanism
LHRH Agonist–Antidepressant	73 (8.94)		
LHRH agonists and TCAs	28 (3.43)	Pharmacodynamic (*n* = 28, 3.43)	Antagonism (*n* = 1, 0.12)
			Additive torsadogenicity (*n* = 27, 3.31)
LHRH agonists and SSRIs	24 (2.94)	Pharmacodynamic (*n* = 24, 2.94)	Additive torsadogenicity (*n* = 24, 2.94)
LHRH agonists and AASRIs	6 (0.74)	Pharmacodynamic (*n* = 6, 0.74)	Additive torsadogenicity (*n* = 6, 0.74)
LHRH agonists and SARIs	6 (0.74)	Pharmacodynamic (*n* = 6, 0.74)	Additive torsadogenicity (*n* = 6, 0.74)
LHRH agonists and NSRIs	3 (0.37)	Pharmacodynamic (*n* = 3, 0.37)	Additive torsadogenicity (*n* = 3, 0.37)
**Others**	**355 (13.60)**		

**Table 11 biomedicines-12-01971-t011:** Torsadogenic risk classification of the selected drugs for this study, according to the QTdrugs List [62].

Known Risk of TdP	Possible Risk of TdP	Conditional Risk of TdP
**PC drugs:**	**PC drugs:** leuprolide and bicalutamide	**PC drugs:** abiraterone and apalutamide
**Psychotropics:** citalopram, escitalopram, chlorpromazine, levomepromazine, mesoridazine, thioridazine, chlorprothixene, haloperidol, droperidol, pimozide, levosulpiride, sulpiride, sultopride, and sertindole	**Psychotropics:** moclobemide, imipramine, trimipramine, nortriptyline, desipramine, maprotiline, venlafaxine, mirtazapine, mianserin, perphenazine, flupentixol, zuclopenthixol, pipamperone, melperone, lumateperone, tiapride, clozapine, asenapine, zotepine, iloperidone, lurasidone, paliperidone, aripiprazole, pimavanserin, lithium, and atomoxetine	**Psychotropics:** amitriptyline, clomipramine, fluoxetine, fluvoxamine, paroxetine, sertraline, trazodone, amisulpride, olanzapine, quetiapine, risperidone, and ziprasidone

## Data Availability

A database containing all DDCs checked for DDIs with their mechanism and severity is present in the Appendix A.

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
