# Peer review of "Concomitant Administration of Psychotropic and Prostate Cancer Drugs: A Pharmacoepidemiologic Study Using Drug–Drug Interaction Databases"

_biomedicines, 2024, doi:10.3390/biomedicines12091971_

Round 1

Reviewer 1 Report

Comments and Suggestions for Authors

The manuscript is well-written but there are some errors in the language in some areas which need to be improved. The methods section need to be explained in more detail. 

Comments on the Quality of English Language

The manuscript is well-written but there are some errors in the language in some areas which need to be improved. 

Reviewer 2 Report

Comments and Suggestions for Authors

The article “Concomitant Administration of Psychotropic and Prostate Cancer Drugs: A Pharmacoepidemiologic Study Using Drug Interaction Databases” is very interesting. I have following comments/suggestions,

1.      The abstract is interesting and clear.

2.      The introduction provides sufficient background into the topic and objectives are clear.

3.      It will be more appropriate to add the dates when authors have accessed the drug interaction databases.

4.      The results are expressed very effectively. Still, Table 4 contains too much information and the authors are advised to divide it into 2 or more tables to make it more interesting for the readers. 

Reviewer 3 Report

Comments and Suggestions for Authors

Dear Editor,

The study, entitled Concomitant Administration of Psychotropic and Prostate Cancer Drugs: A Pharmacoepidemiologic Study Using Drug Inter- 3 action Databases, describes drug-drug interactions with the co-employment of PC drugs and psychotropics, using three drug interaction databases (Lexicomp®, drugs.com®, and Medscape®). The study is sound and it may be efficiently used in clinic as a source.

Some points listed below might increase the quality of the paper.

- Introduction part can be enriched with some sentences on the pathophysiology of prostate cancer.

- A table, on the pharmacological classification drugs used in the treatment of prostate cancer, is missing. 

- Conclusion should also include suggestions and the limitations of the work. 
